# Development of Second-Tier Liquid Chromatography-Tandem Mass Spectrometry Analysis for Expanded Newborn Screening in Japan

**DOI:** 10.3390/ijns7030044

**Published:** 2021-07-14

**Authors:** Yosuke Shigematsu, Miori Yuasa, Nobuyuki Ishige, Hideki Nakajima, Go Tajima

**Affiliations:** 1Department of Pediatrics, Faculty of Medical Sciences, University of Fukui, Fukui 910-1193, Japan; miori@u-fukui.ac.jp; 2Department of Pediatrics, Uji-Tokushukai Medical Center, Uji 611-0041, Japan; 3Division of Newborn Screening, Tokyo Health Service Association, Tokyo 162-8402, Japan; novi.burgi-1579@snow.email.ne.jp; 4Division of Neonatal Screening, Research Institute, National Center for Child Health and Development, Tokyo 157-8535, Japan; nakajima-h@ncchd.go.jp (H.N.); tajima-g@ncchd.go.jp (G.T.)

**Keywords:** isomer, stable-isotope dilution, derivatization, homocystinuria, cobalamin, biotin, maternal 3-methylcronylglycinuria, argininosuccinic acid

## Abstract

To minimize false-positive cases in newborn screening by tandem mass spectrometry in Japan, practical second-tier liquid chromatography-tandem mass spectrometry analyses have been developed using a multimode ODS column with a single set of mobile phases and different gradient elution programs specific to the analysis of acylcarnitines, acylglycines, amino acids, and organic acids. Most analyses were performed using underivatized samples, except for analysis of methylcitric acid, and careful conditioning of the column was necessary for analyses of organic acids. Our second-tier tests enabled us to measure many metabolites useful for detection of target disorders, including allo-isoleucine, homocysteine, methylmalonic acid, and methylcitric acid. We found that accumulation of 3-hydroxyglutaric acid was specific to glutaric acidemia type I and that the ratio of 3-hydroxyisovaleric acid to 3-hydroxyisovalerylcarnitine was useful to detect newborns of mothers with 3-methylcrotonyl-CoA carboxylase deficiency. Data from the analysis of short-chain acylcarnitine and acylglycine were useful for differential diagnosis in cases positive for C5-OH-acylcarnitine or C5-acylcarnitine.

## 1. Introduction

In newborn screening using dried blood spots (DBSs) and flow-injection tandem mass spectrometry (MS/MS) in Japan, a series of acylcarnitines and amino acids, such as valine, leucine (Leu)+isoleucine (Ile), methionine (Met), phenylalanine, and citrulline, have been measured for the screening of fatty acid oxidation disorders, organic acidemias, maple syrup urine disease (MSUD), phenylketonuria, homocystinuria, and citrullinemia, and the recall rates have been relatively high, considering the proposed value [1]. In 2017, 946,065 newborns were born in Japan, and the overall official recall rate was 0.31%, with recall rates of 0.074% for propionylcarnitine (C3), 0.065% for pentanoylcarnitines (C5), 0.009% for hydroxypentanoylcarnitines (C5-OH), 0.055% for pentadioylcarnitines (C5-DC), 0.033% for Leu+Ile, 0.005% for Met, and 0.009% for citrulline. In cases of positive results for screening markers for fatty acid oxidation disorders, serum samples instead of DBSs are analyzed before diagnostic tests such as enzyme assays or gene analyses.

Catabolic conditions in newborns, such as in those who were fed poorly owing to strict breastfeeding, which has been often controversial in Japan, can yield false-positive results in screening for MSUD, glutaric acidemia type I (GA1), carnitine palmitoyltransferase-2 deficiency, and very-long chain acyl-CoA dehydrogenase deficiency in Japan. In addition, false-positive results may be due to low ability to discriminate among isomers of acylcarnitines and amino acids by flow-injection MS/MS. Although allo-isoleucine (allo-Ile) has been reported to be a sensitive disease marker [2,3,4,5] in MSUD screening, allo-Ile values cannot be obtained by flow-injection MS/MS measurement. In GA1, C5-DC is not thought to be a reliable marker for GA1 because the value of C5-DC in flow-injection MS/MS measurement consists of glutarylcarnitine and 3-hydroxyhexanoylcarnitine, both of which are accumulated in a catabolic state, whereas 3-hydroxyglutaric acid (3HGA) in plasma and urine is thought to be a useful diagnostic marker [6,7,8,9].

C3 and C3/C2 are classical screening markers for propionic acidemia (PAE) and a group of methylmalonic acidemias (MMAEs), including defects in cobalamin metabolism and maternal cobalamin deficiency. In the latter disease, C3 values tend to be lower than the traditional cut-offs for PAE and MMAE. We have experienced a false-negative case with cblD who developed MMAE without homocystinuria after viral enteritis in infancy and false-negative cases with cobalamin deficiency, which forced us to adopt lower cut-offs for C3 and C3/C2 and additional screening markers of C3/Met ratio and Met, the latter of which is necessary for methylene tetrahydrofolate reductase deficiency (MTHFRD) screening [10,11]. To detect these disorders, methylmalonic acid (MMA), methyl-citric acid (MCA), 3-hydroxypropionic acid (3HPA), and total homocysteine (tHcy) in DBSs measured by liquid chromatography (LC)-MS/MS have been reported to be powerful disease markers [12,13,14,15,16,17].

C5 is used as a screening marker for isovaleric acidemia (IVAE) and consists of such isomers as isovalerylcarnitine (IVC), valerylcarnitine, 2-methylbutyrylcarnitine (MBC), and pivaloylcarnitine. Positive results for C5 are suggestive of IVAE and 2-methylbutyryl-CoA dehydrogenase deficiency and can occur following the administration of pivaloyl group-containing drugs; moreover, LC-MS/MS analysis of these isomers is useful for differential diagnosis [18,19,20].

Citrulline is a screening marker for argininosuccinic acid synthetase deficiency, argininosuccinic acid lyase deficiency, and citrin deficiency. For the differential diagnosis of these disorders, argininosuccinic acid is measured in DBSs [5].

C5-OH is a screening marker for 3-methylcrotonyl-CoA carboxylase deficiency (3MCCD), multiple carboxylase deficiency, biotin deficiency, 3-hydroxy-3-methylglutaryl-CoA lyase deficiency (HMGLD), and 3-ketothiolase deficiency (KTD), and, in the traditional scheme, additional urinary organic acid analysis using gas chromatography-MS is necessary for differential diagnosis. In these disorders, acylcarnitines, including tiglylcarnitine in plasma, are measured by LC-MS/MS [21], and disease markers, such as tiglylglycine, methylcrotonylglycine (MCG), 3-hydroxyisoveleric acid (HIVA), 3-hydroxy-3-methylglutaric acid (HMGA), and 3-hydroxy-2-methylbutyrylcarnitine (HMBC) in DBSs, are thought to be useful.

Although the above-mentioned LC-MS/MS analyses of the disease markers in DBSs may have promising applications in decreasing recall rates [10,11], different measurement conditions have been reported from various laboratories using different LC columns. In the current study, we developed LC-MS/MS methods to measure many types of marker in DBSs using a single LC column and a single set of mobile phases.

## 2. Materials and Methods

### 2.1. Materials

#### 2.1.1. Biological Samples

DBSs from patients were prepared during the newborn period for mass screening using MS/MS and then stored in a refrigerator in screening laboratories. Samples were then transported to our laboratory at the University of Fukui and measured using LC-MS/MS after obtaining permission from the parents of each patient. Diagnoses for patients with the target disorders in newborn screening were confirmed by enzyme or gene analysis. A patient with tyrosinemia type I was transferred to University of Fukui Hospital because of liver failure at 5 months of age, and succinylacetone levels in DBSs and urine were measured during treatments.

#### 2.1.2. Chemicals

A NeoSMAAT kit for MS/MS newborn screening, which contains labeled acyl-carnitines and amino acids, was purchased from Sekisui Medical Co. (Tokyo, Japan). Methylmalonic acid-d_3_, homocysteine-d_4_, methylcitric acid, methylcitric acid-d_3_, and argininosuccinic acid-^13^C_6_·^15^N_4_ were purchased from Cambridge Isotope Laboratories (Andover, MA, USA); glutaric acid-d_4_, 3-hydroxyglutaric acid, 3-hydroxyglutaric acid-d_4_, and succinylacetone-^13^C_4_ were purchased from the VU Medical Center Metabolic Laboratory (Amsterdam, The Netherlands); 3-hydroxy-3-methylglutaric acid-d_3_, methylcrotonylglycine-d_2_, tiglylglycine-d_2_, and propionylglycine-d_2_ were purchased from CDN Isotope (Point-Claire, Canada); 3-hydroxypropionic acid, 3-hydroxyisovaleric acid, 3-hydroxy-3-methylglutaric acid, pivaloyl chloride, and succinylacetone were purchased from Tokyo Chemical Industry (Tokyo, Japan); 3-hydroxy-2-methylbutyric acid was purchased from Santa Cruz Biotechnology (Dallas, TX, USA); 2-hydroxyglutaric acid was purchased from Toronto Research Chemicals (Toronto, ON, Canada); and allo-Ile was purchased from Wako Chemicals (Kyoto, Japan). Pivaloycarnitine was synthesized in our laboratory using pivaloyl chloride [18].

### 2.2. Methods

LC-MS/MS measurements for metabolites related to screening markers listed in Table 1 were performed.

For FI-MS/MS or LC-MS/MS analysis of acylcarnitines and amino acids, one punch-out (1/8 inch diameter) of a DBS was extracted using 100 μL methanol solution of the NeoSMAAT internal standard kit for routine methods for newborn screening, which contains leucine (5 μM), propionylcarnitine-d_3_ (0.075 μM), isovalerylcarnitine-d_9_ (0.075 μM), and 3-hydroxyisovalerylcarnitine-d_9_ (0.075 μM). After FI-MS/MS analysis, positive samples were analyzed by LC-MS/MS with addition of 2% formic acid water/methanol (1:1) to the plate well.

For analysis of acylcarnitines and acylglycins, one punch-out of a DBS was extracted using 100 µL methanol solution from the NeoSMAAT kit spiked with labeled acylglycines as internal standards: propionylglycine-d_2_ (1.53 μM), tiglylglycine-d_2_ (1.27 μM), and 3-methylcrotonylglycine-d_2_ (1.27 μM).

Total homocysteine and related organic acids were measured according to reported methods [11] with some modifications. The mixture from one punch-out of a DBS and 150 µL reaction solution (acetonitrile/distilled water/formic acid = 59:41:0.42) containing di-thio-threitol (0.77 mg), methylmalonic acid-d_3_ (1.19 μM), homocysteine-d_4_ (0.90 μM), and methylcitric acid-d_3_ (0.69 μM) in a test tube was shaken slowly (120 rpm) at 25 °C for 60 min, and the supernatant was collected after centrifugation. The supernatant was dried under a nitrogen stream and was redissolved in a 2% formic acid water/methanol solution (1:1). For measurement of derivatized samples, the dried extract was derivatized with butanol·HCl at 65 °C for 25 min, dried again, and redissolved in 2% formic acid water/methanol solution (1:1).

For analysis of GA and 3HGA, one punch-out of a DBS was extracted using 100 µL 98% methanol solution containing glutaric acid-d_4_ (0.38 μM) and 3-hydroxyglutaric acid-d_4_ (0.33 μM) as internal standards. For analysis of HIVA, HMGA and HMBA, one punch-out of a DBS was extracted using 100 µL 98% methanol solution containing 3-hydroxy-3-methylglutaric acid-d_3_ (0.30 μM). For analysis of argininosuccinic acid, one punch-out of a DBS was extracted using 100 µL of 90% methanol solution containing argininosuccinic acid-^13^C_6_·^15^N_4_ (1.67 μM). For analysis of succinylacetone, the mixture of one punch-out of a DBS and 110 µL of 80% aceto-nitril solution containing succinylacetone-^13^C_4_ (0.20 μM), 0.1% hydrazine H_2_O, and 0.1% formic acid was stirred slowly for 45 min, and the supernatant was collected after centrifugation. The extract was dried under a nitrogen stream and redissolved in 2% formic acid water/methanol solution (1:1).

The samples (10 μL) were introduced into the LC mobile phase flow (flow rate: 0.4 mL/min) using a 150 mm × 3.0 mm Scherzo SS-C18 column (Imtakt, Portland, OR, USA). Gradient elution of the analytes was achieved using a program with mobile phase A (aqueous 0.5% formic acid) and mobile phase B ((0.5 M ammonium formate/0.5 M NH_4_OH = 9:1)/methanol = 1:9), as detailed in the legends for the corresponding Figures.

For measurements using electrospray-ionization LC-MS/MS, a model API 4000 triple-stage mass spectrometer (AB Sciex, Tokyo, Japan) equipped with a model LC10Avp HPLC system and a model SIL-20AC auto-injector (Shimadzu, Kyoto, Japan) was used [21]. The MS/MS analyses were performed in multiple reaction monitoring (MRM) mode using the transitions detailed in the Figures. Underivatized organic acids were analyzed in negative ion mode. Suitable measurement conditions for the designated transitions were identified with the automatic tune function in Analyst software. For quantification, the recorded peak area of the designated MRM ion set was used.

Allo-Ile was quantified using leucine-d_3_ as an internal standard instead of stable isotope-labeled allo-Ile. The aqueous calibrator for the calibration curve contained 16.7, 83.3, 166.6, or 333.3 μM allo-Ile. Based on the assumption that one punch-out (1/8 inch diameter) of a DBS contains 3 μL whole blood, the mixture of 3 μL of the calibrator and 100 μL internal standard solution from the NeoSMAAT internal standard kit was analyzed to determine the linearity. In the mixture, the concentration of allo-Ile was 0.1×, 0.5×, 1×, or 2× that of leucine-d_3._

Similarly, pivaloylcarnitine was quantified using isovalerylcarnitine-d_9_ as an internal standard instead of stable isotope-labeled pivaloylcarnitine. The calibrator for the calibration curve contained 0.25, 1.25, 2.5, or 5.0 μM pivaloylcarnitine, and the mixture of 3 μL the calibrator and 100 μL the internal standard solution from the NeoSMAAT internal standard kit was analyzed.

HIVA and HMBA were quantified using 3-hydroxy-3-methylglutaric acid-d_3_ as an internal standard. The calibrator for calibration curve contained 1.0, 5.0, 10.0, or 20.0 μM HMGA, HIVA, and HMBA, and the mixture of 3 μL calibrator and 100 μL internal standard solution was analyzed.

To determine the linearity of analyses other than allo-Ile, pivaloylcarnitine, HIVA and HMBA, we also analyzed the mixture of calibrator and internal standard, in which the target metabolite level was 0.1×, 0.5×, 1×, or 2× that of the internal standard. Intra- and inter-assay imprecisions were tested by analysis of patient DBSs.

## 3. Results

In analyses using the stable isotope dilution method with the stable isotope-labeled compound as an internal standard, the calibration curves were linear in the test concentration range. Intra- and inter-assay CV values in analyses of patient DBSs were less than 10%, except for those of HIVA and HMBA assays.

An LC-MS/MS chromatogram of the allo-Ile measurement using a DBS from a newborn with MSUD is shown in Figure 1. Allo-Ile concentrations in DBSs were calculated based on a calibration curve using the aqueous solutions of allo-Ile and leucine-d_3_, which was linear (R^2^ = 0.9994) up to an allo-Ile concentration corresponding to 333.3 μM in DBS. Those of patients with MSUD are listed in Table 2, together with those in false-positive cases. Intra-assay CV (n = 5) and inter-assay CV (n = 5) were 5.4% and 8.1%, respectively, at an allo-Ile concentration of 38.3 μM in DBS.

LC-MS/MS chromatograms for analyses of MMA, MCA, and tHcy in DBSs from a newborn with cobalamin deficiency type C (cblC), using an underivatized sample (A) and a derivatized sample (B), are shown in Figure 2. In our analysis of underivatized samples, the peaks of MCA did not show good quality, although this was obtained for the analysis of derivatized samples. In the analysis of derivatized samples, 3HPA concentrations could not be measured well, likely owing to the low ionization efficiency of 3HPA-butylester and sample loss during preparation.

In derivatized sample measurements from control newborns (n = 13), the concentrations of MMA, MCA, and tHcy ranged from 0.20 to 0.99 μM, 0.20 (below the detection limit) to 0.75 μM, and 1.1 to 4.9 μM, respectively. In patients with MMAE, including cblA and cblD (n = 8), MMA concentrations ranged from 15.0 to 863.9 μM. MCA concentrations obtained from derivatized sample measurements in patients with PAE (n = 13) ranged from 0.92 to 3.50 μM, whereas 3HPA concentrations obtained from underivatized sample measurements in patients with PAE (n = 13) and control newborns ranged from 9.6 to 32.8 μM and 1.7 to 8.8 μM, respectively.

In underivatized sample measurements, MMA and tHcy concentrations in patients with cblC, maternal cobalamin-deficiency, MFHFRD, and CBSD are shown in Table 3. These patients were characterized by elevated tHcy concentrations.

LC-MS/MS chromatogram for analysis of GA and 3HGA in the DBS from a newborn with GA1 is shown in Figure 3. 3HGA was quantified based on a transition that was different from that of 2-hydroxyglutaric acid. The concentrations of 3HGA in newborn DBSs from three patients ranged from 1.08 to 1.44 μM (mean ± standard deviation in controls: 0.35 ± 0.10), and those of glutaric acid (GA) ranged from 12.1 to 25.8 μM (in controls: 6.67 ± 2.95).

LC-MS/MS chromatograms for analyses of HMG, HIVA, and HMBA in the DBS of a newborn with KTD are shown in Figure 4. The concentrations of HIVA and HMBA were calculated based on the calibration curves using the aqueous solutions of HIVA, HMBA and HMGA-d_3_. The calibration curves were linear, and intra- and inter-assay CV values in analyses of patient DBSs are given in Table 4.

LC-MS/MS chromatograms for analyses of acylcarnitines and acylglycines in DBSs from a newborn with KTD (a) and a newborn with 3MCCD (b) are shown in Figure 5. In KTD, increased tiglylcarnitine, tiglylglycine, and 3-hydroxy-2-methylbutyrylcarnitine levels were observed, whereas increased 3-methylcrotonylglycine and 3-hydroxyisovalerylcarnitine (HIVC) levels were characteristic in 3MCCD. The MRM transition of *m*/*z* 262 > 145, in addition to that of 262 > 85, was used for measurement of 3-hydroxyisovalerylcarnitine, since complete chromatographic separation between 3-hydroxyisovalerylcarnitine and 3-hydroxy-2-methylbutyrylcarnitine (HMBC) was not achieved [21].

Metabolites in patients with diseases characterized by high C5-OH-acylcarnitine concentrations are listed in Table 5. 3-Ketothiolase deficiency was characterized by elevated 3-hydroxy-2-methylbutyric acid (HMBA), 3-hydroxy-2-methylbutyrylcarnitine (HMBC), and tiglylglycine; HMGLD was characterized by elevated HMGA; holo-carboxylase deficiency was characterized by elevated HIVA, HIVC, and propionylglycine; and 3MCCD was characterized by elevated HIVA, HIVC, and 3-methylcrotonylglycine. In babies born to mothers with 3MCCD, the ratios of HIVA to HIVC (0.2–2.9) were markedly lower than those (23.8, 79.9) in patients with 3MCCD.

LC-MS/MS chromatogram for analysis of short-chain acylcarnitines in the DBS from a newborn treated with antibiotics is shown in Figure 6. The peaks of 3 isomers appeared separately, and the condition was characterized by increased pivaloylcarnitine concentrations, which were calculated based on a calibration curve using aqueous solutions of pivaloylcarnitine and isovalerylcarnitine-d_9_. The calibration curve was linear (R^2^ = 0.9995) up to the pivaloylcarnitine concentration corresponding to 5.5 μM in DBS. Intra-assay CV (n = 5) and inter-assay CV (n = 5) for pivaloylcarnitine were 4.2% and 9.2%, respectively, at a pivaloylcarnitine concentration of 3.7 μM in DBS. Concentrations of pivaloylcarnitine and IVC in control newborns (n = 13) were below the detection limit (0.01 μM) and 0.17 ± 0.10 μM, respectively. Those of pivaloylcarnitine in newborns treated with antibiotics ranged 1.2 to 9.7 μM, and those of IVC in patients with isovaleric acidemia ranged from 1.5 to 17.2 μM.

LC-MS/MS chromatograms for analyses of argininosuccinic acid in DBSs of a newborn with argininesuccinate lyase deficiency and a control newborn are shown in Figure 7. The limit of quantification was 0.05 nmol/mL in DBSs.

LC-MS/MS chromatogram for analysis of succinylacetone in the DBS from a patient with tyrosinemia type 1 is shown in Figure 8, together with the clinical course and succinylacetone concentrations for the patient. The succinylacetone concentration in the newborn DBS stored in a refrigerator for 5 months was 28.7 nmol/mL, whereas that in control newborn DBSs was 0.21 ± 0.10 nmol/mL.

## 4. Discussion

To manage positive cases with screening markers such as C3, C5, C5-OH, C5-DC, Leu+Ile, Met, and citrulline, we developed practical second-tier tests using a single LC column and a single set of mobile phases together with different gradient elution programs specific for the designated LC-MS/MS measurements. We used a multimode ODS Scherzo SS-C18 column with anion and cation exchange and showed excellent chromatographic ability for amino acids, short- to medium-chain acylcarnitines, and acylglycines. In addition, our methods for acylcarnitines and allo-Ile analysis were convenient because the positive samples could be measured in wells of a plate following addition of 2% formic acid/methanol for our second-tier LC-MS/MS.

Despite these advancements, LC-MS/MS measurements of organic acids are still challenging. Organic acids with multiple carboxyl groups are difficult to analyze using underivatized samples [13], and analytical methods for derivatized samples have been adopted in several laboratories [11,14]. However, analysis of butylated organic acids may still be difficult owing to the relatively poor ionization efficiency in electrospray-ionization or the high volatility of small molecule organic acids. Because MS/MS measurements of organic acids are performed in negative mode, whereas those of amino acids are performed in positive mode, MS/MS instruments with the ability to switch quickly between the two modes are needed for measurement of both organic acids and amino acids when using underivatized samples. Moreover, some stable isotope-labeled organic acids are not available from reagent manufacturers.

In the current study, using underivatized samples, MCA could not be quantified, and 3HPA was measured using methylmalonic acid-d_3_ as an internal standard because stable isotope-labeled 3HPA was not available. Nevertheless, 3HPA levels in our underivatized sample measurements were found to be useful for practical detection of mild PAE. Next, we used methods for sample preparation and derivatization for analyses of MMA, MCA, and tHcy [11], which enabled us to achieve practical LC-MS/MS analysis using our mobile phases for the SS-C18 column. MCA levels obtained with this measurement approach were useful for detection of PAE.

Preferably, LC-MS/MS analysis should be performed using underivatized samples because time-consuming sample preparation processes may result in damage to analytes. In addition, derivatization can be challenging because of the need for a fume hood and drying apparatus in screening laboratories. LC-MS/MS measurement of tHcy together with MMA is useful for screening of a series of homo-cystinurias [11]. In our system, measurement of tHcy and MMA could be performed using both derivatized and underivatized samples. Indeed, our results for tHcy and MMA levels provided additional useful information in the screening of cblC, cobalamin deficiency, MTHFRD, and CBSD. In Japan, a pilot study of homocystinuria screening with modified cut-offs for C3 and C3/C2 and an additional marker of C3/Met is currently underway, combined with a second-tier test for MMA and tHcy measurement using underivatized samples.

Our LC-MS/MS measurements were based mostly on the stable isotope dilution technique. However, suitable stable isotope-labeled internal standards were not available for the quantification of some acylcarnitines, including tiglylcarnitine, pivaloylcarnitine, 2-methylbutyrylcarnitine, and 3-hydroxy-2-methylbutyrylcarnitine, although the values for these acylcarnitines, calculated using isovalerylcarnitine-d_9_ as an internal standard, are practically precise in newborn screening. In contrast, measurements of organic acids, such as HIVA and 3HMBA, using 3-hydroxy-3-methylglutaric acid-d_3_ as an internal standard, should be performed with careful conditioning of the column in order to obtain precise values. Notably, the ratio of HIVA to HIVC may be useful for identifying babies born to mothers with 3MCCD because the practice of identifying mothers with mild 3MCCD using elevated C5-OH in DBSs of newborns may be controversial.

Regarding LC-MS/MS measurement of organic acids, 3HGA appears to have an important role when screening for GA1. Although DBS levels of glutarylcarnitine and GA apparently overlap between patients with GA1 and control newborns, those of 3HGA in patients with GA1 were significantly higher than those in control newborns. The wide distribution of glutarylcarnitine and GA levels may be affected by catabolic conditions in control newborns.

For practical application of newborn screening projects, identification of patients as early as possible is thought to be essential for initiation of appropriate treatment based on the laboratory data specific to the disease. Thus, data measured by LC-MS/MS for the follow-up of patients have been obtained from several screening laboratories [3,8,17,22]. Our methods can be applied to serum sample measurements, and serum and DBS concentrations of metabolites, such as MMA, tHcy, 3HGA, allo-Ile, argininosuccinic acid, and succinylacetone, by LC-MS/MS have been provided to hospitals for patient follow-up from our laboratories. Argininosuccinic acid measurement was sufficiently sensitive in our measurements compared with previously reported methods [5] and was useful to discriminate argininesuccinate lyase deficiency from argininesuccinate synthetase deficiency and citrin deficiency, while the screening kits that allow us to discriminate these disorders are not used, since argininesuccinate lyase deficiency is quite rare in Japan. Moreover, argininosuccinic acid data are used for evaluating the effects of long-term treatment. Tyrosinemia type I is extremely rare and is not included in the list of target disorders for newborn screening in Japan, and succinylacetone data obtained using LC-MS/MS may be used for the follow-up of patients, as shown in Figure 8.

Unfortunately, our second-tier tests have not yet been used in most of the screening laboratories in Japan. In Japan, 872,683 babies were born in 2020, and samples from newborns were tested in 37 screening laboratories. In the majority of these laboratories, fewer than 10,000 newborns are tested annually using a single LC-MS/MS instrument, and LC-MS/MS measurements as second-tier tests seemed to be a significant burden to a limited number of staff, mainly because of the additional work required to maintain equipment performance, despite our simple measurement approach. Consolidation of screening work in a reduced number of laboratories and an additional LC-MS/MS instrument for second-tier tests, with some type of kit for quality assurance, including sufficient labeled internal standards, may facilitate the use of these tests in screening laboratories.

## Figures and Tables

**Figure 1 IJNS-07-00044-f001:**
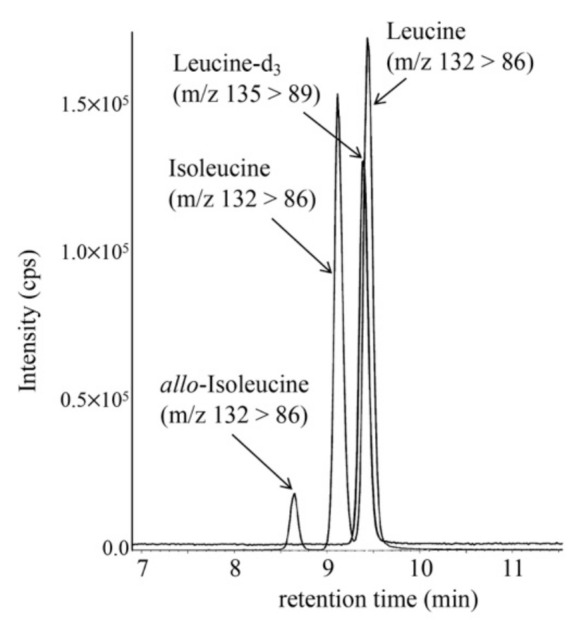
LC-MS/MS analysis of allo-Ile was performed using the following gradient elution program: 20% B (0.1 min), 20% to 30% B (3 min), 30% to 51% B (7 min), 51% to 100% B (0.1 min), 100% B (7.9 min), 100% to 20% B (0.1 min), and 20% B (5 min), with a flow rate of 0.3 mL/min.

**Figure 2 IJNS-07-00044-f002:**
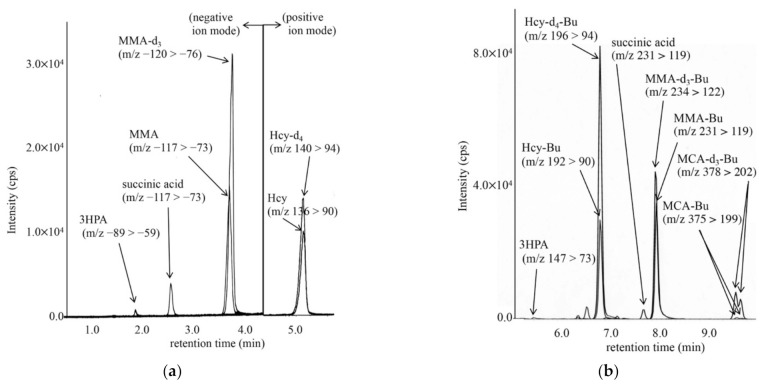
LC-MS/MS analysis of underivatized organic acids, including methylmalonic acid (MMA) using negative ion mode and homocysteine (Hcy) by positive ion mode (**a**), and that for derivatized organic acids, including methylcitric acid butyl-ester (MCA-Bu) and Hcy butyl-ester (Hcy-Bu) by positive ion mode (**b**). The gradient elution program in (**a**) was as follows: 13% B (0.5 min), 13% to 30% B (6 min), 30% to 100% B (0.1 min), 100% B (7.9 min), 100% to 10% B (0.1 min), and 13% B (5 min); that in (**b**) was as follows: 0% B (0.2 min), 0% to 80% B (1 min), 80% to 100% B (9 min), 100% B (3 min), 100% to 0% B (0.1 min), and 0% B (3 min).

**Figure 3 IJNS-07-00044-f003:**
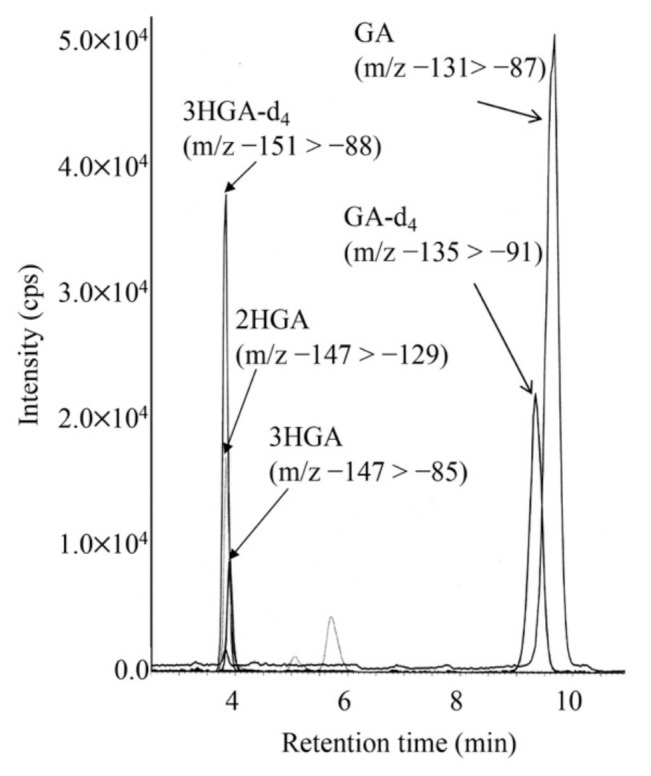
LC-MS/MS analysis for underivatized glutaric acid (GA), 3-hydroxyglutaric acid (3HGA), and 2-hydroxyglutaric acid (2HGA) using negative ion mode. The gradient elution program was as follows: 0% B (5 min), 0% to 30% B (6 min), 30% to 100% B (0.1 min), 100% B (7.9 min), 100% to 10% B (0.1 min), and 13% B (5 min).

**Figure 4 IJNS-07-00044-f004:**
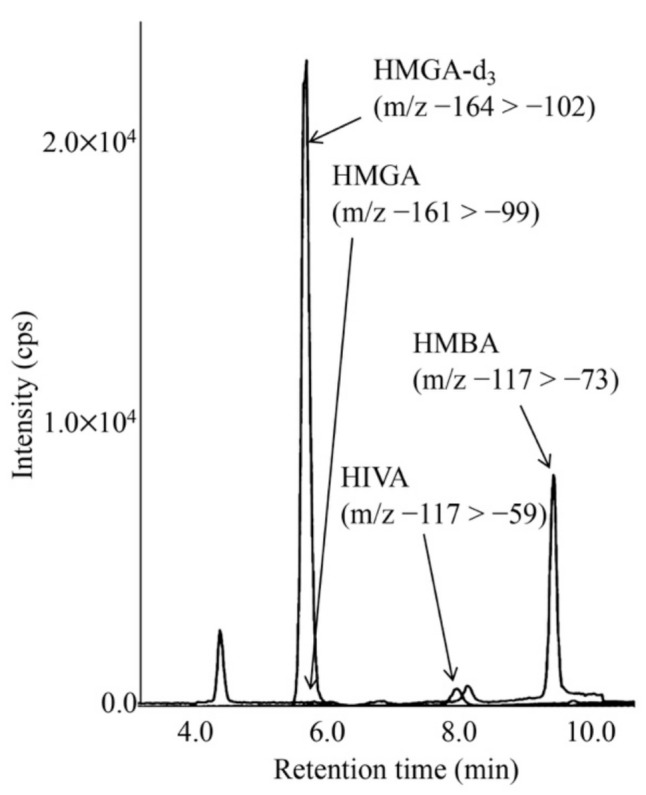
LC-MS/MS analysis for underivatized 3-hydroxy-3-methylglutaric acid (HMGA) and 3-hydroxyisovaleric acid (HIVA) using negative ion mode. The gradient elution program was as follows: 0% B (5 min), 0% to 38% B (6 min), 38% to 100% B (0.1 min), 100% B (5.9 min), 100% to 0% B (0.1 min), and 0% B (5 min).

**Figure 5 IJNS-07-00044-f005:**
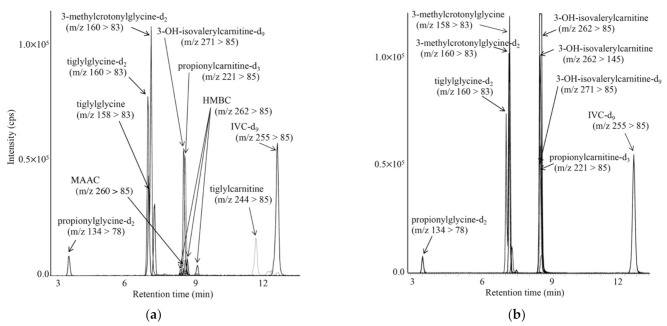
LC-MS/MS analyses for propionylglycine, tiglylglycine, and 3-methylcrotonylglycine, together with acylcarnitines, including tiglylcarnitine, using positive ion mode for a newborn with KTD (**a**) and a newborn with 3MCCD (**b**). The gradient elution program was as follows: 10% B (1 min), 10% to 40% B (4 min), 40% to 59% B (9 min), 55% to 100% B (0.1 min), 100% B (6.9 min), 100% to 10% B (0.1 min), and 10% B (5 min).

**Figure 6 IJNS-07-00044-f006:**
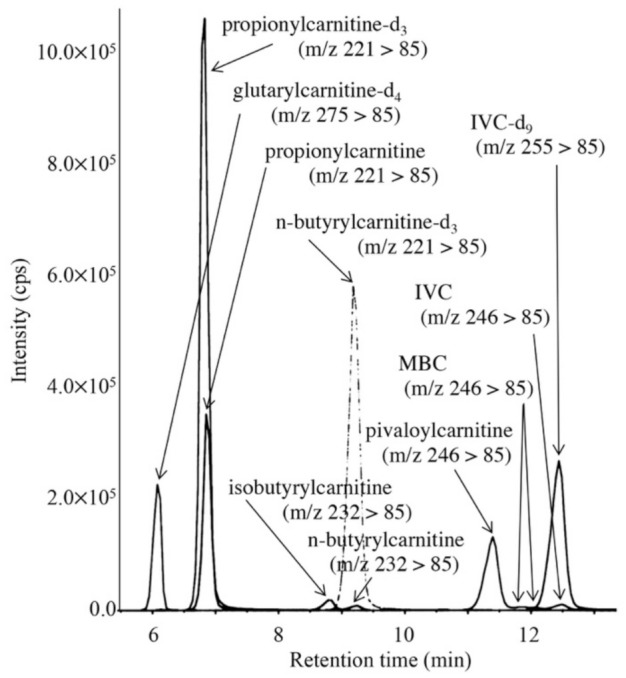
LC-MS/MS analysis for pivaloylcarnitine, 2-methylbutyrylcarnitine, isovalerylcarnitine, n-butyrylcarnitine, and isobutyrylcarnitine. The gradient elution program was as follows: 10% B (1 min), 10% to 40% B (4 min), 40% to 70% B (8 min), 70% to 100% B (0.1 min), 100% B (6.9 min), and 100% to 10% B (0.1 min).

**Figure 7 IJNS-07-00044-f007:**
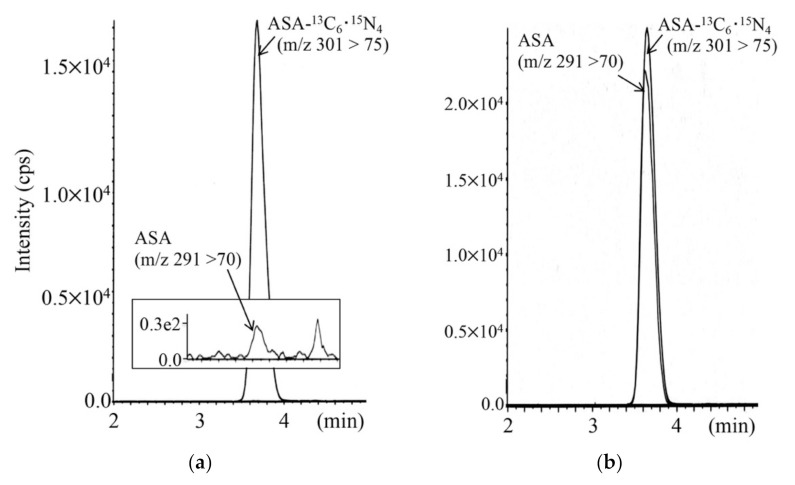
LC-MS/MS analysis for argininosuccinic acid (ASA) using positive ion mode for a control newborn (**a**) and a newborn with argininesuccinate lyase deficiency (**b**). The gradient elution program was as follows: 0% B (1 min), 0% to 30% B (4 min), 30% to 100% B (0.5 min), 100% B (5.5 min), 100% to 10% B (0.5 min), and 10% B (5 min).

**Figure 8 IJNS-07-00044-f008:**
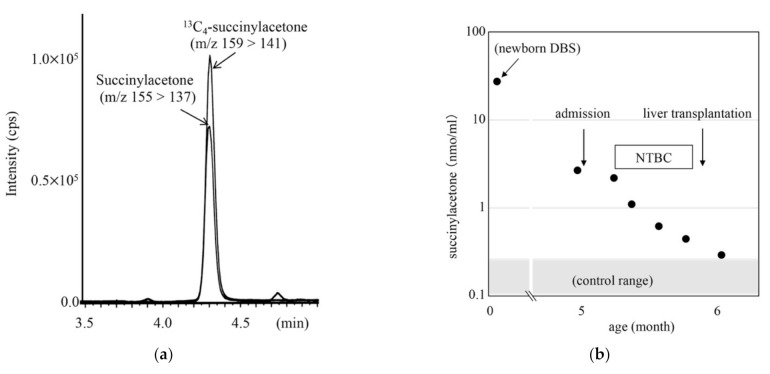
LC-MS/MS analysis for succinylacetone (SA) using positive ion mode (**a**) and clinical course of a patient with tyrosinemia type 1 (**b**). Gradient elution program: 20% B (0.1 min), 20% to 100% B (5 min), 100% B (5 min), 100% to 20% B (0.5 min), and 20% B (4 min).

**Table 1 IJNS-07-00044-t001:** LC-MS/MS methods for the metabolites related to the screening markers.

LC-MS/MS Method	Positive Screening Marker	Target Metabolite
1	Leu+Ile	allo-Ile, Ile, Leu
2-A	C3, C3/C2, Met	MMA, 3HPA, tHcy
2-B	C3, C3/C2, Met	MMA, MCA, tHcy (derivatized)
3	C5-DC	GA, 3HGA
4	C5-OH, C5:1	HIVA, HMGA, HMBA
5	C5-OH, C5:1	short-chain acylcarnitines, acylglycines
6	C5	short-chain acylcarnitines
7	citrulline	argininosuccinic acid

**Table 2 IJNS-07-00044-t002:** Allo-Ile, related amino acid, and acetylcarnitine levels (μM) in patients with maple syrup urine disease, as measured by LC-MS/MS.

MSUD Patient #	FI-MS/MS			LC-MS/MS
	Acetylcarnitine	Valine	Leu+Ile	Allo-Ile
1	40.7	368	767	38.3
2	20.5	427	554	82.1
3	15.0	386	2199	362.6
4	17.9	508	813	131.0
false positive cases (n = 12)	15.0–38.2	211–420	304–420	0.5–5.9

**Table 3 IJNS-07-00044-t003:** Methylmalonic acid (MMA) and total homocysteine (tHcy) in DBSs of patients with positive results in screening markers of C3 (propionylcarnitine), C3/C2, C3/Met, and Met (methionine).

Pt #	Diagnosis	FI-MS/MS				LC-MS/MS	
		C3 (μM)	C3/C2	C3/Met	Met (μM)	MMA (μM)	tHcy (μM)
1	cblC	10.30	1.10	1.60	6.40	59.7	34.8
2	cblC	15.01	1.10	0.46	32.5	44.4	17.0
3	cobalamin deficiency ^1^	3.59	0.21	0.34	10.65	6.7	5.5
4	cobalamin deficiency ^1^	2.14	0.45	0.35	6.12	5.5	11.8
5	MTHFRD	0.62	0.13	0.13	4.98	0.5	49.2
6	MTHFRD	0.45	0.05	0.07	6.63	0.8	10.6
7	MTHFRD	0.77	0.09	0.17	4.61	1.3	10.4
8	MTHFRD	0.53	0.09	0.08	6.70	0.8	28.7
9	CBSD ^2^	1.00	0.02	0.00	911	0.1	84.7
upper cut-off		3.5	0.25	0.25	80	1.0	5.0
lower cut-off		-	-	-	9.27	-	-

^1^ The infants with cobalamin-deficiency were born to mothers with ileum-resection and chronic atrophic gastritis, and were exclusively breast-fed for several months and developed symptoms. The values were obtained using their DBSs for newborn screening which were stored in refrigerators. ^2^ cystathionine beta-synthase deficiency.

**Table 4 IJNS-07-00044-t004:** Assay validation for HMGA, HIVA, and HMBA analysis.

Analyte	Linearity (R^2^)	Imprecision		
		Analyte Level in DBS (μM)	CV (%) Intraassay (n = 5)	CV (%) Inter-assay (n = 5)
HMGA	0.9994	5.1	5.1	7.9
HIVA	0.9982	14.7	7.6	12.2
HMBA	0.9974	39.1	8.9	14.4

**Table 5 IJNS-07-00044-t005:** Organic acids, acylcarnitines, and acylglycines in DBSs from newborns with high C5-OH levels.

Diagnosis	FI-MS/MS	LC-MS/MS										
	C5-OH (µM)	Organic Acid (µM)			Acylcarnitine (µM)					Acylglycine (µM)		
		HMGA	HIVA	3HMBA	Propionyl	HIVC	HMBC	Tiglyl	MCC ^1^	Propionyl	Tiglyl	MCG ^2^
3-ketothiolase deficiency	3.1	0.37	2.2	130.3	0.67	0.12	4.84	0.49	<0.01	0.02	2.67	0.01
	2.8	0.10	1.3	39.1	1.62	0.13	3.61	0.62	<0.01	0.09	4.10	<0.01
HMGLD	3.1	5.19	25.3	0.66	0.19	2.78	0.01	0.01	<0.01	0.04	0.01	0.07
Holo-carboxylase deficiency	2.2	1.04	624.5	0.44	1.78	1.44	0.02	0.03	<0.01	1.49	0.26	2.10
	3.4	0.19	187.5	0.53	4.12	3.35	0.01	<0.01	0.01	0.73	0.02	0.91
mild biotin deficiency	1.1	0.04	5.8	0.11	0.81	1.09	0.01	<0.01	<0.01	0.01	0.01	0.04
3MCCD	11.9	0.23	238.5	0.60	0.13	10.01	0.01	<0.01	<0.01	0.01	0.01	3.66
	3.4	0.45	326.6	0.47	0.85	4.09	0.01	<0.01	0.01	0.03	0.01	1.08
Baby born to mother with 3MCCD	6.8	0.11	14.7	0.39	0.41	5.10	0.01	<0.01	<0.01	0.02	0.03	0.43
	3.9	0.24	11.7	0.55	0.86	5.69	0.01	<0.01	<0.01	0.02	0.02	0.01
	4.8	0.06	1.0	0.20	0.39	4.54	0.06	<0.01	<0.01	0.01	0.01	0.01
controls (mean ± SD)	<0.5	0.53 ± 0.20	2.1 ± 0.6	0.60 ± 0.11	1.17 ± 0.45	0.09 ± 0.03	<0.01	<0.01	<0.01	0.02 ± 0.01	0.01 ± 0.01	<0.01

^1^ 3-methylcrotonylcarnitine, ^2^ 3-methylcrotonylglycine.

## Data Availability

The data used to support the findings of this study are available from the corresponding author upon request.

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
