# Peer review of "Development of Second-Tier Liquid Chromatography-Tandem Mass Spectrometry Analysis for Expanded Newborn Screening in Japan"

_2409-515X, 2021, doi:10.3390/ijns7030044_

Round 1

Reviewer 1 Report

The authors report the development of second tier-test methods on a single column, in order to measure many metabolites useful for detection of target disorders.

The work is quite interesting; however, some clarifications are needed.

  • Lines 161-162. The authors state that the biomarker values are obtained from a calibration curve, performed using the relative stable isotope-labeled compound: what is the dynamic and / or linearity range? Assuming that the calibration curve has also been used in the other methods, it would be appropriate to report the linearity ranges for each methods.

  • Table 2 shows low allo-isoleucine values for false positives. However, the values of valine and leu+Ile are not so high for them. What are the screening cut-off values?

  • In Table 3, “maternal cobalamin-deficiency” could be specified, as reported in line 202

  • Table 3 describes the data of patients affected by CblC and maternal cobalamin deficiency. In lines 57-58 the authors state that “C3 values tend to be lower than the traditional cut-offs for PAE and MMAE, and we have experienced false-negative cases with cblD and cobalamin deficiency”. So, how does CblD correlate with MMA? Does CblD show values as high as those of CblC or are they lower as in maternal deficit?

  • Lines 261-263. The authors state that “holocarboxylase deficiency was characterized by elevated HIVA, HIVC, and propionylglycine; and 3MCCD was characterized by elevated HIVA, HIVC, and 3-methylcrotonylglycine”, however data reported in Table 4 disclose elevated MCG in holocarboxylase deficiency.

  • Lines 275-276. The authors report the values of pivaloylcarnitine and IVC in the controls, which were the values in the patients?

  • Lines 371-374. The authors state that the second tier test for argininsuccinic acid is useful to discriminate argininosucinate lyase deficiency from argininosucinate synthetase deficiency. There are commercial kits that allow you to discriminate the two diseases on the first screening test because they include the measurement of arginisuccinic acid. Have you thought about using them?

  • Line 128: “acylglycines” instead of “acylglycies”

  • Lines 43-46. The authors state that catabolic conditions in newborns can yield false-positive results in screening for some diseases, in Japan. Why is it specified in Japan? Are there evidences of a high number of false positives in Japan rather than in other countries?

Author Response

(Reviewer’s comments)

The authors report the development of second tier-test methods on a single column, in order to measure many metabolites useful for detection of target disorders.

The work is quite interesting; however, some clarifications are needed.

    Lines 161-162. The authors state that the biomarker values are obtained from a calibration curve, performed using the relative stable isotope-labeled compound: what is the dynamic and / or linearity range? Assuming that the calibration curve has also been used in the other methods, it would be appropriate to report the linearity ranges for each method.

Response: We added the concentrations of internal standards in IS solutions and the methods to test the linearity ranges and intra- and interassay imprecisions in the Methods section. Then, we added data regarding linearity and CV to the Results section.

(Reviewer’s comments)

    Table 2 shows low allo-isoleucine values for false positives. However, the values of valine and leu+Ile are not so high for them. What are the screening cut-off values?

Response: DBSs of false-positive cases were collected from several screening laboratories. In most screening laboratories in Japan, cut-off values were set based on the 99.9th percentile value in control newborns and varied among the laboratories. Thus, the values of valine or leu+Ile in false-positive cases were judged as higher than cut-off values in each laboratory.

(Reviewer’s comments)

    In Table 3, “maternal cobalamin-deficiency” could be specified, as reported in line 202

Response: In Table 3, the following sentences were added: “Infants with cobalamin deficiency were born to mothers with ileum resection or chronic atrophic gastritis and were exclusively breast-fed for several months and developed symptoms. The values were obtained from their DBSs for newborn screening, which were stored in refrigerators.”

(Reviewer’s comments)

    Table 3 describes the data of patients affected by CblC and maternal cobalamin deficiency. In lines 57-58 the authors state that “C3 values tend to be lower than the traditional cut-offs for PAE and MMAE, and we have experienced false-negative cases with cblD and cobalamin deficiency”. So, how does CblD correlate with MMA? Does CblD show values as high as those of CblC or are they lower as in maternal deficit?

Response: CblD regulates the biosynthesis and proportions of two coenzymes, i.e., methylcobalamin and 5'-deoxyadenosylcobalamin, and dysfunction of this protein results in MMAE or MMAE with homocystinuria. We experienced a false-negative case with cblD, in which the patient developed MMAE without homocystinuria after viral enteritis in infancy. The C3 and C3/C2 values in newborn DBSs were lower than the traditional cut-off values of 3.5 mM and 0.25, respectively; data were not shown in Table 3 because the patient did not have homocystinuria.

We have revised the text (lines 59–61) in the Introduction as follows:

We experienced a false-negative case with cblD in a patient who developed MMAE without homocystinuria after viral enteritis in infancy and false-negative cases with cobalamin deficiency.

(Reviewer’s comments)

    Lines 261-263. The authors state that “holocarboxylase deficiency was characterized by elevated HIVA, HIVC, and propionylglycine; and 3MCCD was characterized by elevated HIVA, HIVC, and 3-methylcrotonylglycine”, however data reported in Table 4 disclose elevated MCG in holocarboxylase deficiency.

Response: Elevated MCG was observed both in holocarboxylase deficiency and 3MCCD and was not specific to holocarboxylase deficiency. Thus, we did not list elevated MCG as a characteristic marker.

(Reviewer’s comments)

    Lines 275-276. The authors report the values of pivaloylcarnitine and IVC in the controls, which were the values in the patients?

Response: We added a sentence to describe these data (lines 300–304) as follows:

Concentrations of pivaloylcarnitine and IVC in control newborns (n = 13) were below the detection limit (0.01 mM) and 0.17 ± 0.10 mM, respectively. Those of pivaloylcarnitine in newborns treated with antibiotics ranged from 1.2 to 9.7 mM, and those of IVC in patients with isovaleric acidemia ranged from 1.5 to 17.2 mM.

(Reviewer’s comments)

    Lines 371-374. The authors state that the second tier test for argininosuccinic acid is useful to discriminate argininosucinate lyase deficiency from argininosucinate synthetase deficiency. There are commercial kits that allow you to discriminate the two diseases on the first screening test because they include the measurement of argininosuccinic acid. Have you thought about using them?

Response: In Japan, argininosucinate lyase deficiency is rare, and kits that enable discrimination between the two diseases on initial screening are not used. Treatments for the two diseases during the newborn period are similar, and it is thought that our second-tier test may be useful in Japan.

Practically, argininosuccinic acid could be measured using FI-MS/MS with the kits without stable isotope-labeled argininosuccinic acid and Sciex API4000 in our laboratory, although it may be difficult to measure this compound in most screening laboratories in Japan.

We revised the text (lines 399–403) in the Discussion section as follows:

Argininosuccinic acid measurement was sufficiently sensitive in our measurements compared with previously reported methods [5] and was useful to discriminate argininosuccinate lyase deficiency from argininosuccinatesynthetase deficiency and citrin deficiency; screening kits enabling discrimination among these disorders are not commonly used because argininosuccinate lyase deficiency is quite rare in Japan.

(Reviewer’s comments)

    Line 128: “acylglycines” instead of “acylglycies”

Response: The typo was corrected.

Reviewer 2 Report

The manuscript nicely presents a set of methods for second-tier analysis. The methods are described without details about their performance (e.g loq, CV% etc.) but the focus is rather on their usefulness.

The title of the manuscript implies that it will describe how second-tier methods are in used in expanded newborn screening in Japan, but as I understand from the manuscript this is not the case. The manuscript describes a set of methods that has been developed, but as I can understand from the manuscript they are not used in the screening labs. I therefore think that the title needs to be changed to better match the content of the manuscript, i.e. that it describes a (set of) developed method(s).

In the last paragraph of the introduction, the content of the study is presented, as consisting of two aims: 1. We developed LC-MS/MS methods to measure many types of markers in DBSs using a single LC column and a single set of mobile phases. 2. We then examined whether our measurement approach was useful and practical for second-tier tests in screening laboratories in Japan.
The first aim, the developed methods are well described, but the second aim is hardly mentioned at all, only in the last paragraph of the discussion. I don't think this is enough to say that the authors has "examined whether our measurement approach was useful and practical...". Therefore, unless an examination of the kind that is mentioned is carried out (for example letting one of the screening labs set up and use the methods as a second tier for the screening and evaluate its usefulness), I think that the last paragraph of the introduction needs to be changed.

The preparation of calibrators is not mentioned, I suggest that this is shortly mentioned in the materials and methods section (from the results sections I can read that for at least some of the methods aqueous solutions are used).

P.10, row 314-416: The procedure described here (a dilution/resuspension of the first-tier sample) should be described in the materials and methods section as well.

Typos:
Keywords and p.3 row 142: arginiosuccinic acid should be argininosuccinic acid

Author Response

(Reviewer’s comments)

The manuscript nicely presents a set of methods for second-tier analysis. The methods are described without details about their performance (e.g loq, CV% etc.) but the focus is rather on their usefulness.

The title of the manuscript implies that it will describe how second-tier methods are in used in expanded newborn screening in Japan, but as I understand from the manuscript this is not the case. The manuscript describes a set of methods that has been developed, but as I can understand from the manuscript they are not used in the screening labs. I therefore think that the title needs to be changed to better match the content of the manuscript, i.e. that it describes a (set of) developed method(s).

Response: Based on the reviewer's comments, the title has been changed to the following: “Development of Second-tier Liquid Chromatography-tandem Mass Spectrometry Analysis for Expanded Newborn Screening in Japan”.

(Reviewer’s comments)

In the last paragraph of the introduction, the content of the study is presented, as consisting of two aims: 1. We developed LC-MS/MS methods to measure many types of markers in DBSs using a single LC column and a single set of mobile phases. 2. We then examined whether our measurement approach was useful and practical for second-tier tests in screening laboratories in Japan.

The first aim, the developed methods are well described, but the second aim is hardly mentioned at all, only in the last paragraph of the discussion. I don't think this is enough to say that the authors has "examined whether our measurement approach was useful and practical...". Therefore, unless an examination of the kind that is mentioned is carried out (for example letting one of the screening labs set up and use the methods as a second tier for the screening and evaluate its usefulness), I think that the last paragraph of the introduction needs to be changed.

Response: Based on the reviewer's comments, the last sentence in the last paragraph of the Introduction (lines 90–92) has been changed as follows:

In the current study, we developed LC-MS/MS methods to measure many types of markers in DBSs using a single LC column and a single set of mobile phases. We then examined whether our measurement approach could be acceptable for use in screening laboratories in Japan.

In addition, the last paragraph of the Discussion was changed as follows:

Unfortunately, our second-tier tests have not yet been used in most screening laboratories in Japan. In Japan, 872,683 babies were born in 2020, and samples from newborns were tested in 37 screening laboratories. In the majority of these laboratories, fewer than 10,000 newborns are tested annually using a single LC-MS/MS instrument, and LC-MS/MS measurements as second-tier tests seem to be a significant burden to the limited number of staff, mainly because of the additional work required to maintain equipment performance despite our simple measurement approach. Consolidation of screening work in a reduced number of laboratories and an additional LC-MS/MS instrument for second-tier tests with some type of kit for quality assurance may facilitate the use of these tests in screening laboratories.

(Reviewer’s comments)

The preparation of calibrators is not mentioned, I suggest that this is shortly mentioned in the materials and methods section (from the results sections I can read that for at least some of the methods aqueous solutions are used).

Response: We added a description of the preparation of calibrators in the Materials and Methods section.

(Reviewer’s comments)

P.10, row 314-416: The procedure described here (a dilution/resuspension of the first-tier sample) should be described in the materials and methods section as well.

Response: We added a sentence regarding the measurement of the first-tier sample to the revised text (lines 131–133) as follows:

After FI-MS/MS analysis, positive samples were analyzed by LC-MS/MS with addition of 2% formic acid water/methanol into the plate well.

(Reviewer’s comments)

Typos:

Keywords and p.3 row 142: arginiosuccinic acid should be argininosuccinic acid

Response: The typos were corrected. 

Round 2

Reviewer 2 Report

As a reply to my previous comment “The preparation of calibrators is not mentioned, I suggest that this is shortly mentioned in the materials and methods section (from the results sections I can read that for at least some of the methods aqueous solutions are used).”, you responded that “We added a description of the preparation of calibrators in the Materials and Methods section.”. However, I cannot find that this has been added. Please add it, including information on the calibrator ranges for the different analytes.

p.4, row 173 says: “To determine the linearity of each analysis, the internal standard solution, which was spiked with the compound at concentrations of 2x, 1x, 0.5x, or 0.1x of the internal standard, was measured”. I don’t understand this. Does it mean that the calibrators have concentrations ranging from 0,1-2x the concentration of the internal standards? This does not make sense. You say that allo-Ile is linear to a concentration of 324 µM (row 184). The IS concentration (5 µM) x2 is 10 µM.

p4. Row 179: “Intra- and inter-assay CV values in analyses of patient DBSs were less than 10%.” This is not correct. On the end of p.7 the interassay CV for HIVA is given as 12,2 %, och for HMBA as 14,4%. I suggest that you change this sentence into “Intra- and inter-assay CV values in analyses of patient DBSs are given in table …” and put these in a table instead. The linear ranges can preferably be placed in the table as well.

In Materials and Methods, concentrations are given in nmol/mL, while in the results section, the concentrations are given in µM. Please use the same unit throughout the paper.

At several occasions in the Materials and Methods section, “2% formic acid water/methanol” is used. At one occation it says “2% formic acid water/methanol solution (1:1). Is this what is meant all the time? Please specify.

p.4, row 157: “…was collected after centrifuged” should be “…was collected after centrifugation”

According to my previous comment:
In the last paragraph of the introduction, the content of the study is presented, as consisting of two aims: 1. We developed LC-MS/MS methods to measure many types of markers in DBSs using a single LC column and a single set of mobile phases. 2. We then examined whether our measurement approach was useful and practical for second-tier tests in screening laboratories in Japan.

The first aim, the developed methods are well described, but the second aim is hardly mentioned at all, only in the last paragraph of the discussion. I don't think this is enough to say that the authors has "examined whether our measurement approach was useful and practical...". Therefore, unless an examination of the kind that is mentioned is carried out (for example letting one of the screening labs set up and use the methods as a second tier for the screening and evaluate its usefulness), I think that the last paragraph of the introduction needs to be changed.

Response: Based on the reviewer's comments, the last sentence in the last paragraph of the Introduction (lines 90–92) has been changed as follows:

In the current study, we developed LC-MS/MS methods to measure many types of markers in DBSs using a single LC column and a single set of mobile phases. We then examined whether our measurement approach could be acceptable for use in screening laboratories in Japan.

In addition, the last paragraph of the Discussion was changed as follows:

Unfortunately, our second-tier tests have not yet been used in most screening laboratories in Japan. In Japan, 872,683 babies were born in 2020, and samples from newborns were tested in 37 screening laboratories. In the majority of these laboratories, fewer than 10,000 newborns are tested annually using a single LC-MS/MS instrument, and LC-MS/MS measurements as second-tier tests seem to be a significant burden to the limited number of staff, mainly because of the additional work required to maintain equipment performance despite our simple measurement approach. Consolidation of screening work in a reduced number of laboratories and an additional LC-MS/MS instrument for second-tier tests with some type of kit for quality assurance may facilitate the use of these tests in screening laboratories.

Reviewers answer to the response:

The clarification in the discussion is good. In the introduction however, please remove: “We then examined whether our measurement approach could be acceptable for use in screening laboratories in Japan.” This study does not cover that topic (there are no method or results regarding this).  

Author Response

According to the additional comments from Reviewer 2, we have revised our article as follows:

(Reviewer’s comments)

As a reply to my previous comment “The preparation of calibrators is not mentioned, I suggest that this is shortly mentioned in the materials and methods section (from the results sections I can read that for at least some of the methods aqueous solutions are used).”, you responded that “We added a description of the preparation of calibrators in the Materials and Methods section.”. However, I cannot find that this has been added. Please add it, including information on the calibrator ranges for the different analytes.

p.4, row 173 says: “To determine the linearity of each analysis, the internal standard solution, which was spiked with the compound at concentrations of 2x, 1x, 0.5x, or 0.1x of the internal standard, was measured”. I don’t understand this. Does it mean that the calibrators have concentrations ranging from 0,1-2x the concentration of the internal standards? This does not make sense. You say that allo-Ile is linear to a concentration of 324 µM (row 184). The IS concentration (5 µM) x2 is 10 µM.

Response: We appreciate these comments from the reviewer. We have added text to the revised manuscript to explain the calibrators used in this study (linens 184 - 200), as follows:

 Allo-isoleucine was quantified using leucine-d3 as an internal standard instead of stable isotope-labeled allo-isoleucine. The aqueous calibrator for the calibration curve contained 33.3, 83.3, 166.6, or 333.3 µM allo-isoleucine. Based on the assumption that one punch-out (1/8 inch diameter) of a DBS contains 3 µL whole blood, the mixture of 3 µL of the calibrator and 100 µL internal standard solution from the NeoSMAAT internal standard kit was analyzed to determine the linearity. In the mixture, the concentration of allo-isoleucine was 0.1×, 0.5×, 1×, or 2× that of leucine-d3.

Similarly, pivaloycarnitine was quantified using isovalerylcarnitine-d9 as an internal standard instead of stable isotope-labeled pivaloycarnitine. The calibrator for the calibration curve contained 0.25, 1.25, 2.5, or 5.0 µM pivaloycarnitine, and the mixture of 3 µL the calibrator and 100 µL the internal standard solution from the NeoSMAAT internal standard kit was analyzed.

HIVA and HMBA were quantified using 3-hydroxy-3-methylglutaric acid-d3 as an internal standard. The calibrator for calibration curve contained 1.0, 5.0, 10.0, or 20.0 µM HMGA, HIVA, and HMBA, and the mixture of 3 µL calibrator and 100 µL internal standard solution was analyzed.

To determine the linearity of analyses other than allo-isoleucine, pivaloylcarnitine, HIVA and HMBA, we also analyzed the mixture of calibrator and internal standard, in which the target metabolite level was 0.1×, 0.5×, 1×, or 2× that of the internal standard. Intra- and inter-assay imprecisions were tested by analysis of patient DBSs.

(Reviewer’s comments)

p4. Row 179: “Intra- and inter-assay CV values in analyses of patient DBSs were less than 10%.” This is not correct. On the end of p.7 the interassay CV for HIVA is given as 12,2 %, och for HMBA as 14,4%. I suggest that you change this sentence into “Intra- and inter-assay CV values in analyses of patient DBSs are given in table …” and put these in a table instead. The linear ranges can preferably be placed in the table as well.

Response: We appreciate this comment from the reviewer. Based on the reviewer's comments, we changed the sentence, and added Table 4 (“Assay validation for HMGA, HIVA and HMBA analysis”; lines 308 - 313) as follows:

The calibration curves were linear, and intra- and interassay CV values in analyses of patient DBSs are given in Table 4.

Table 4 Assay validation for HMGA, HIVA, and HMBA analysis

Analyte

Linearity (R2)

Imprecision

Analyte level in DBS (µM)

CV (%) intraassay (n=5)

CV (%) interassay (n=5)

HMGA

0.9994

5.1

5.1

7.9

HIVA

0.9982

14.7

7.6

12.2

HMBA

0.9974

39.1

8.9

14.4

Additionally, the first paragraph in the Result was corrected as follows:

In analyses using the stable isotope dilution method with the stable isotope-labeled compound as an internal standard, the calibration curves were linear in the test concentration range. Intra- and interassay CV values in analyses of patient DBSs were less than 10%, except for those of HIVA and HMBA assays.

(Reviewer’s comments)

In Materials and Methods, concentrations are given in nmol/mL, while in the results section, the concentrations are given in µM. Please use the same unit throughout the paper.

Response: We thank the reviewer for this comment. Accordingly, we changed nmol/mL to µM.

(Reviewer’s comments)

At several occasions in the Materials and Methods section, “2% formic acid water/methanol” is used. At one occation it says “2% formic acid water/methanol solution (1:1). Is this what is meant all the time? Please specify.

Response: We thank the reviewer for this comment. The text “2% formic acid water/methanol solution (1:1)” is correct. We changed the phrase “2% formic acid water/methanol solution” to “2% formic acid water/methanol solution (1:1)” throughout the paper.

(Reviewer’s comments)

p.4, row 157: “…was collected after centrifuged” should be “…was collected after centrifugation”

Response: We appreciate this comment from the reviewer. The typos were corrected.

(Reviewer’s comments)

The clarification in the discussion is good. In the introduction however, please remove: “We then examined whether our measurement approach could be acceptable for use in screening laboratories in Japan.” This study does not cover that topic (there are no method or results regarding this). 

Response: We thank the reviewer for this comment. Accordingly, we removed the last sentence in the Introduction.

Round 3

Reviewer 2 Report

Here comes one last comment:

Row 174-175: “The aqueous calibrator for the calibration curve contained 33.3, 83.3, 166.6, or 333.3 mM allo-isoleucine.” This does not fit with the concentrations 0.1×, 0.5×, 1×, or 2× that of leucine-d3. Instead it should probably be “…16.7, 83.3, 166.6, or 333.3 mM allo-isoleucine”

Author Response

(Reviewer’s comments)

Row 174-175: “The aqueous calibrator for the calibration curve contained 33.3, 83.3, 166.6, or 333.3 mM allo-isoleucine.” This does not fit with the concentrations 0.1×, 0.5×, 1×, or 2× that of leucine-d3. Instead it should probably be “…16.7, 83.3, 166.6, or 333.3 mM allo-isoleucine”

Response: We appreciate the comment from the reviewer. We changed the sentence as follows:

 The aqueous calibrator for the calibration curve contained 16.7, 83.3, 166.6, or 333.3 mM allo-isoleucine.
